

# CNOP based on ACPW for Identifying Sensitive Regions of Typhoon Target Observations with WRF Model

Bin Mu[1], Linlin Zhang[1], Shijin Yuan[1], Wansuo Duan[2]

5  [1]School of Software Engineering, Tongji University, Shanghai 201804
[2]State Key Laboratory of Numerical Modeling for Atmospheric Sciences and Geophysical Fluid Dynamics, Institute of Atmospheric Physics, Chinese Academy of Sciences,Beijing 100029, China

*Correspondence to:* Shijin Yuan (yuanshijin2003@163.com)

**Abstract.**

In this paper, we rewrite the ACPW (adaptive cooperation co-evolution of parallel particle swarm optimization and wolf search algorithm based on principal component analysis) and applied it to solve conditional nonlinear optimal perturbation (CNOP) in the WRF-ARW for identifying sensitive areas of typhoon target observations, which is proposed by us in the study of Zhang

et al. (2018), to investigate its feasibility and effectiveness in the WRF-ARW model. Fitow (2013) and Matmo (2014) are taken as two typhoon cases, and simulated with the 60 km horizontal resolution. The total dry energy is adopted as the objective function. The CNOP is also calculated by the method based on the adjoint model (ADJ-method) as a benchmark. To evaluate the ACPW-CNOP, five aspects are analysed, such as the pattern, energy, similarity, benefits from the CNOPs reduced in the whole domain and the sensitive regions identified, and the simulated typhoon tracks. The experimental results show that the

temperature and wind patterns of ACPW-CNOP is similar to those of the ADJ-CNOP in all typhoons. And the similarity values of ADJ-CNOP and ACPW-CNOP of two typhoon cases are more than 0.5. When reducing CNOPs in the sensitive regions, the forecast income of ACPW-CNOP is greater than that of ADJ-CNOP in all typhoons. Moreover, the sensitive regions identified by the ACPW-CNOP has the similar influence with the ADJ-CNOP on the simulation of typhoon tracks, sometimes the ACPW-CNOP has more positive impact on the simulation of typhoon tracks. The ACPW is more efficient than the ADJ-

method in this paper.

## 1 Introduction

Target observation is an effective way to improve the typhoon forecast skills, which is a study of the typhoon predictability (Franklin and Demaria, 1992; Bergot, 1999; Aberson, 2003). Conditional nonlinear optimal perturbation (CNOP) proposed by Mu and Duan (2003) is an effective method for studying the typhoon predictability (Mu and Duan, 2003). And many

researchers use CNOP method to identify sensitive areas of typhoon target observations (TTOs). Most of the current studies of sensitive areas identification adopt the MM5 (The Fifth-Generation Mesoscale Model) model (Zhou and Mu, 2011; Zhou



and Mu, 2012a; Zhou and Mu, 2012b; Zhou and Zhang, 2014). Zhou and Mu (2011) investigate the affection of the different verification regions to the sensitive area identifications, and summary that the little shift range and size of the verification regions will not affect the structure of CNOP. Zhou and Mu (2012a) also study the affection of different horizontal resolutions and found that the higher resolution will show the more small-scale information of CNOP. They also research the different

dependency of integration times and regimes (2012b), and the solutions guide the future research. Zhou and Zhang (2014) study three sensitive area identification schemes and recommend the vertically integrated energy scheme.

However, the MM5 model is not updated and maintained after 2006. Nowadays, the popular mesoscale model is WRF-ARW (Advanced Research the Weather Research and Forecast) model. Recently, there is only one study to identify sensitive areas by using the WRF-ARW model (Skamarock et al, 2008). Yu et al. (2017) use the spectral projection gradient 2 (SPG2)

algorithm (Ernesto et al., 2001) to solve CNOP, but the adjoint model of WRF-ARW only has one gravity dragging boundary layer parameterization scheme for such study, which limits the simulation of typhoon. In addition, when the horizontal resolution is higher than 30km, the gradient information calculated by the adjoint model has errors and omissions, which results in falling into the local optimum or optimization failure. Hence, a method without using adjoint model is needed.

Wen et al. (2014) proposed a modified intelligent algorithm (IAs) called SAEP (simulated annealing-based ensemble

projecting method) to solve CNOP in the Zebiak-Cane (ZC) (Zebiak and Cane, 1987) model for studying the ENSO predictions. PCGD (principal components-based great deluge) (Wen et al., 2015a), RGA (robust PCA-based genetic algorithm) (Wen et al., 2015b), CTS-SS (continuous Tabu search algorithm with sine maps and staged strategy) (Yuan et al., 2015), and PCAGA (principal component analysis-based genetic algorithm) (Mu et al., 2015b) also were proposed to do the same study. These methods were useful and effective. But it should be investigated that whether these methods can be used to solve CNOP in the

MM5 model and WRF-ARW model for identifying sensitive areas of TTOs.

Therefore, we adopted the PCAGA to solve CNOP in the MM5 model with the 120 km horizontal resolution. The experiments showed that PCAGA method was free of the adjoint model and also obtained CNOPs which had similar and meaningful physical patterns with the benchmark (adjoint-based method's results). In addition, the CNOPs obtained by PCAGA method have more positive influences over the forecast improvement than by adjoint-based method. However, the PCAGA was not

paralleled and its performance is worse than the PPSO in the ZC model. Hence, we combined the advantages of particle swarm optimization (PSO) and wolf search algorithm (WSA), and proposed a new modified IA, ACPW (adaptive cooperation co-evolution of parallel particle swarm optimization and wolf search algorithm based on principal component analysis) (Zhang et al., 2018). The ACPW was used to solve CNOP in the MM5 model with two horizontal resolutions, 60 km and 120 km. We compared the CNOP achieved by the ACPW with ADJ-CNOP, and the experimental results showed that the ACPW-CNOP

had the similar patterns, high similarity, the higher benefits, faster run time and the same influence on the typhoon tracks simulation. This type methods have not been applied to the WRF-ARW yet.

Hence, in this paper, we rewrite the ACPW method and applied it to solve CNOP in the WRF-ARW for identifying sensitive areas of typhoons adaptive observations. We take two typhoons as study cases, Fitow (2013) and Matmo (2014), and simulate them with the 60 km horizontal resolution. Similar to our previous study (Zhang et al., 2018), the total dry energy is adopted



as the objective function. To evaluate the CNOPs from the ACPW method, we compare them with the benchmark in terms of the patterns, energies, similarities and benefits from the CNOPs reduced in the whole domain and sensitive regions, and the simulated typhoon tracks. All experimental results show that in the WRF-ARW model the ACPW method also is feasible and effective for solving CNOPs to identify the sensitive regions of TTOs.

The rest of the paper is organized as follows. The brief description of CNOP and ACPW method is denoted in the section 2. Section 3 and section 4 are the parts of experiments, whose design is in the section 3 and analysis and results are in the section 4. The last part about the conclusions is in section 5.

## 2 CNOP and ACPW

### 2.1 CNOP

CNOP is an initial perturbation $\zeta\varphi_0^*$ of vector $\Phi_0$ (initial basic state) under the constrain condition $\|\varphi_0\|^2 \leq \zeta$, if and only if

$$\begin{cases} J(\zeta\varphi_0^*) = \max\limits_{\|\varphi_0\|^2 \leq \zeta} J(\varphi_{NT}) \\ \varphi_{NT} = PM(\Phi_0 + \zeta\varphi_0) - PM(\Phi_0) \end{cases}, \qquad (1)$$

where P is a local projection operator with setting 1 inside of the verification region and 0 outside, and the verification region is a key area considered by researchers, which is represented in Fig. 1.

Figure 1 shows the schematic diagram of verification region, which is denoted by the red square, and when model has more than one vertical level, verification region of each level is the same. In addition, different cases have different verification regions.

$$\Phi_t = M_{t_0 \to t}(\Phi_0), \qquad (2)$$

M denotes a nonlinear model and $\Phi_t$ is the propagator of M from the initial time $t_0$ to the predicted time t.

We convert the objective function $J(\zeta\varphi_0^*)$ to a problem of seeking minimum.

$$J(\zeta\varphi_0^*) = \min\limits_{\|\varphi_0\|^2 \leq \zeta} -J(\varphi_{NT}), \qquad (3)$$

### 2.2 ACPW

The ACPW method was proposed by Zhang et al. (2018), which is used to solve CNOP in the MM5 model for identifying the sensitive regions of TTOs. The ACPW has two points, one is the cooperation co-evolution of PSO and WSA, the other is the

two adaptive subswarms. The details and pseudocode of ACPW is described in Table 1. And the control parameters of ACPW are list in Table 2.

The elaborations of the PSO and WSA update rules were in the study of Zhang et al. in 2018 (Zhang et al., 2018).

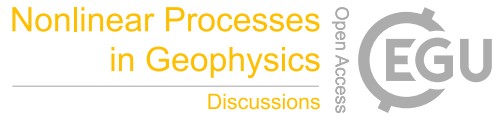

## 3 Experimental Design

All the experiments are run on a Lenove Thinkserver RD430 with two Intel Xeon E5-2450 2.10 GHz CPUs, 32 logical cores and 132G RAM. And the operating system is CentOS 6.5. All the codes are written in FORTRAN language and compiled by PGI Compiler 10.2.

### 3.1 The model and Data

In this paper, we adopt the WRF-ARW model and its corresponding adjoint system (Zhang et al., 2013) to study the sensitive areas identification of TTOs. We produce the initial and boundary conditions by using the FNL (Final Analysis) data from NCEP (National Centers for Environmental Prediction) (NCEP, 2000) at 1°×1° and 6-h intervals. The physical parameterization schemes of the WRF-ARW are constructed as dry convective adjustment, the surface drag planetary boundary

layer scheme, grid resolved large-scale precipitation and the Kuo cumulus parameterization scheme. We also use observed typhoon tracks (Ying et al., 2014) from the China Meteorological Administration (CMA) - Shanghai Typhoon Research Institute to evaluate the simulated typhoon tracks of the WRF-ARW model.

### 3.2 Typhoons Fitow (2013) and Matmo (2014)

We take two typhoons as the study cases, Fitow (2013) and Matom (2014). Fitow is the 23st typhoon of 2013, and develop on

September 29 to the east of Philippines. On October 6, Fitow strikes China at Fuding in Fujian province, with a landfall pressure of 955 hPa. Typhoon Matmo (2014) is the 10th typhoon of 2014. It initially happened on July 17 and made landfall in Taiwan on July 22. In these two cases, a set of 24-h control forecasts, which served as basic state, are investigated from 0000 UTC 5 Oct 2013 to 0000 UTC 6 Oct 2013 (Fitow), and from 1800 UTC 21 Jul 2014 to 1800 UTC 22 Jul 2014 (Matom). During these 24 hours, the maximum sustained winds up to 45 meters per second for typhoon Fitow and 42 meters per second

for Matmo, before the typhoons hits land. For each case, the forecast is executed at a 60-km resolution with 21 vertical levels with the top pressure at 50hPa, and the model domain covers 55×55 grids.

The simulated typhoon tracks of WRF-ARW model are presented in Fig. 2. The ability of the WRF-ARW model simulating these cases accurately are checked built on a 24-h simulation initialized at 0000 UTC 5 Oct 2013 and 1800 UTC 21 Jul 2014. Figure 2(a) shows the model simulated track of typhoon Fitow (hollow) runs a little faster than the observed track (solid) and

25 moves to the south after 0012 UTC 05 Oct 2013. Figure 2(b) shows the model simulated track of typhoon Matmo (hollow) moves along the observed track (solid) but slightly faster, and after 6 hours has a little migration. All these errors are acceptable in this study.

### 3.3 Experimental setup

As the conducted physical parameterization schemes, we only focus on the dry physical process in this paper. Therefore, the

30 initial perturbation $\zeta\varphi_0$ related to four dry physical characteristics, i.e., the perturbed zonal wind $u_0'$, meridional wind $v_0'$,





temperature $T'_0$ and surface pressure $p'_{s0}$. The objective function is calculated by the total dry energy (Zhou and Zhang, 2014) in formula (4).

$$J(\varphi_{NT}) = \frac{1}{D}\int_D \int_0^1 \frac{1}{2}\left(u'^2_t + v'^2_t + \frac{c_p}{T_r}T'^2_t\right)d\eta dD + \frac{1}{D}\int_D R_a T_r \left(\frac{p'_{st}}{p_r}\right)^2 dD, \qquad (4)$$

where $u'_t$, $v'_t$, $T'_t$, $p'_{st}$ are components of $\varphi_{NT}$, which is the nonlinear development of perturbed $\Phi_0$ (i.e. $\Phi_0 + \zeta\varphi_0$) from the initial time $t_0$ to the prediction time $t$. $\eta$ is the vertical coordinate. $D$ is the verification area. And other reference parameters with constant values are shown in Table 3.

For optimizing conveniently, the formula of solving CNOP in formula (3) can be transformed to solve a minimization problem, as follows:

$$J(\zeta\varphi_0^*) = \min_{\|\varphi_0\|^2 \le \zeta}\left(-\frac{1}{D}\int_D \int_0^1 \frac{1}{2}\left(u'^2_t + v'^2_t + \frac{c_p}{T_r}T'^2_t\right)d\eta dD + \frac{1}{D}\int_D R_a T_r \left(\frac{p'_{st}}{p_r}\right)^2 dD\right), \qquad (5)$$

## 4. Experimental Results and Analysis

To verify the feasibility and validity of the ACPW in the WRF-ARW model, we compare CNOPs obtained with those from the ADJ-method in terms of the pattern, energy, similarity, benefits from the CNOPs reduced in the whole domain and sensitive regions, as well as the simulated typhoon tracks.

### 4.1 CNOP pattern

The pattern is the most important standard among the evaluation standards for CNOP. The patterns of ADJ-CNOP and ACPW-CNOP of the WRF-ARW are denoted in the Figures 3 and 4. Figure 3 is for typhoon Fitow, and typhoon Matmo is shown in Figure 4. The shaded parts represent the temperature and the vectors describe the wind. The patterns are the vertical level at $\eta = 0.7$, i.e. the 500 hPa atmospheric layer, which is focused on by researchers generally. From the figures, we can find that the patterns of ACPW-CNOP is similar to those of the ADJ-CNOP in all typhoons. The distribution of warm and cold temperature zones is approximately the same, and the direction of wind vector is almost the same. Except for the CNOP patterns of ACPW are more dispersed and fragmented than those of ADJ-method.

As we use the total vertical dry energy to identify the sensitive regions of typhoons, the distribution of the vertical dry energy is presented in Figure 5 and Figure 6. And the figures show the area with the first 1.2% energy.

For typhoon Fitow, the energy almost has the same position, which is distributed in the north side of the verification area. The difference is the ACPW-CNOP has another secondary part in the southeast side of the verification area.

However, the energy position of ACPW-CNOP is different with the ADJ-CNOP for typhoon Matmo in Figure 6. The energy distribution of these two CNOPs are concentrated in two blocks, but the deviation of the position is large. The energy of ADJ-CNOP is mainly distributed in the southwest and east side of the verification area. One major part in the east side is crescent-



shaped with large power, and the other one in the southwest has smaller energy than the former. The energy of ACPW-CNOP distributes in the northwest and northeast of the verification area, which has the similar power.

To sum it up, the distribution of CNOP pattern is more similar then that of the CNOP energy, and under the condition of the 60-km resolution, the similarity of upon distribution is higher than those in the MM5 model of the paper published by Zhang
et al. in 2018 (Zhang et al., 2018).

In order to further analyze the similarity and the forecasting benefit of the identified sensitive region, the following numerical analysis experiments are carried out in this paper.

### 4.2 Numerical similarity

The numerical similarity between ACPW-CNOP and ADJ-CNOP is calculated by formula (6).

$$S_{xy} = \frac{\langle X,Y \rangle}{\sqrt{\langle X,X \rangle}\sqrt{\langle Y,Y \rangle}}, \tag{6}$$

X and Y represent the vectors of CNOPs obtained from the ACPW and ADJ-method. The similarity values are list in Table 4. 0.61 is the similarity value of the two CNOPs of Typhoon Fitow, and 0.53 is for typhoon Matmo. Compare to the pattern similarity in Figures 3 and 4, the numerical similarity is smaller. It is because that the pattern is plotted by the values of one
vertical level, while the numerical similarity is calculated by the all values of CNOP including all physical quantities and vertical levels. Even so, the similarity values also are more than 0.5.

### 4.3 Benefits from reduction of CNOPs

The experiments of this section include two parts: the forecasting benefits obtained by reducing CNOP to W × CNOP in the whole domain, i.e. the CNOP values of all grid points are reduced; the forecasting benefits from CNOP to W × CNOP is
reduced only in the sensitive regions, i.e. the CNOP values of the sensitive grid points are reduced to 0.75 × CNOP, 0.5 × CNOP and 0.25 × CNOP.

All experiments are based on two assumptions that:

a. When adding target observations in the identified sensitive areas, the environment around is idealized, and the improvements of observations added are reducing original errors to 0.75, 0.5 and 0.25 times.

b. CNOPs achieved by us can be seen as the optimal initial perturbations. Once we reduce them in the sensitive regions, the benefits earned will be the best.

As mentioned in the subsection 4.1, the sensitive region is determined according to the first 1.2% of total vertical dry energy, as shown in Figures 5 and 6, which are the shadow zones in the figures.



### 4.3.1 Reducing CNOP to W × CNOPs in the whole domain

Figure 7 shows the forecasting benefits of reducing CNOP to W × CNOPs in the whole domain of ADJ-CNOP and ADJ-CNOP.

From Figure 7, we can see that the forecasting benefits of ACPW-CNOP is greater than that of ADJ-CNOP in typhoon Fitow,
while the result is opposite in typhoon Matmo.

### 4.3.2 Reducing CNOP to W × CNOPs in the sensitive regions

The sensitive regions of TTOs are identified by the upon strategy for selecting points, which are plotted in the Figures 5 and
6. The forecasting benefits are list in Table 5.

From the numerical results in Table 5, it can be seen that the forecast income of ACPW-CNOP in the sensitive region is greater
than that of ADJ-CNOP in all typhoons. Especially, the benefits of ACPW-CNOP are much larger than those of ADJ-CNOP,
and the gap is two orders of magnitude. In addition, when the W coefficient of ADJ-CNOP decreases gradually, the forecast
income of ADJ-CNOP keeps basically unchanged, about 6.9% in typhoon Fitow and about 0.08% in typhoon Matmo. While
the forecast benefit of ACPW-CNOP changes obviously. The benefit value of typhoon Fitow is 10.2871%, 9.6823%, 8.8120%
respectively, and typhoon Matmo is 2.3484%, 3.5716% and 5.2212% respectively.

## 4.4 Efficiency analysis

As the efficiency analysis of solving CNOP in the MM5 model (Zhang et al., 2018), the time consumption belongs to the times
of calling the non-linear model. From the experiments, the time consumption of the non-linear model of WRF-ARW is more
than that of the MM5. Hence, we must consider the influence from the values of the control parameters of the ACPW, such as
the number of principle components $n'$, the number of individuals N and the number of iterations Total_Step.

In this subsection, we set several group values of upon three control parameters to investigate, and the other parameters are
the same to Table 2. The running time is list in Table 6 under the condition of running WRF-ARW with 32 logical cores
parallelly. In addition, the running time is the average value of four experiments under the same conditions. The average
running time of ADJ method is 232.31 minutes, and the total running time of using four initial first guess fields is 929.24
minutes. All the results are based on the Fitow case. The Matmo case has similar results.

Eight representative control parameters and the experimental results are list in Table 6. From the experimental analysis, the
number of principal components, i.e. selected dimensions of the feature space, has little effect on the running time, but has
great influence on the adaptive value of objective function.

When the number of individuals and iteration steps remain unchanged but the dimension setting increases gradually, the
adaptive value of objective function of ACPW-CNOP decreases gradually, referring to lines 5 and 7 of Table 6. This is due to
the increasing dimension needs more individual and iterations to optimize in the feature space, if not it will resulte in the
method falling into a local optimum.





It also can be seen that the running time of ACPW is proportional to the number of $N \times Total\_Step$. When the number of $N \times Total\_Step$ is the same, the running time of ACPW is almost the same. When the number of $N \times Total\_Step$ is increased, the running time also increases proportionally. Moreover, from line 1 to line 3 of the values, it is found that the increasing individuals does not increase the adaptive values. From line 2 and line 4 of the values, the iterations is increased, the adaptive

values increase. We can conclude that the increasing iterations is beneficial to the optimization of the ACPW.

Observing the adaptive values, it is found that the values of line 4 and line 6 are more than 30000, and the corresponding running time is about 360 minutes. When we use the parameters of line 1, the adaptive value is 28126.185933, which is little smaller than 30000, but its running time is only 89.83 minutes, and the benefits are in Figures 7. Therefore, we can set some small number of the control parameters to do the experiments. When using the parameters in line 1 and the ADJ-method using

one first guess field, the speedup of the ACPW is 2.59. If the ADJ-method uses four first guess fields, the speedup of the ACPW is 10.34. In this paper, the ACPW is more efficient than the ADJ-method.

## 4.5 Simulation of the typhoon tracks

In order to investigate the validity of CNOP in identifying sensitive regions, we compare the 24-hour simulated typhoon track

by adding CNOP or W × CNOP to the initial states. Similar to the benefits, there are two ways to modify the CNOP value: one is to reduce the CNOP value to 0.75, 0.5 and 0.25 times in the whole domain; the other is to reduce the CNOP value to 0.5 times only in the sensitive regions of TTOs.

In order to show the effect clearly, only two tracks are drawn in each sub-figure, one is the best typhoon track provided by CMA, and the other is the simulation track of the WRF-ARW model with different CNOPs, as shown in Figures 8 and 9. Since

the difference of typhoon tracks simulated by the WRF-ARW model after adding modified CNOPs is very small, it is difficult to find them when they are displayed in the figures. Therefore, the experimental results of this part are also shown in Tables 7 and 8 in a numerical way. The larger values in the tables are longitude (E) and the smaller values are latitude (N).

### 4.5.1 Simulated track analysis of adding CNOP modified in the whole domain

Firstly, we analyse the simulated Fitow tracks of the WRF-ARW model after adding modified ADJ-CNOP in the whole domain

to the initial state, and the results are shown in Figures 8 and 9.

Combining with the tracks in Figure 8 and the data in Table 7, it can be seen that the track of typhoon Fitow has little difference, which is simulated by the WRF-ARW model after reducing the value of CNOP in the whole domain and adding it on the initial state. One difference is the position of simulated typhoon tracks at 1200 UTC 5 Oct. Only the position with adding the whole ADJ-CNOP is different with that of adding another W×ADJ-CNOPs. The second difference is the position of 0.5×CNOP and

0.25×CNOP at 1800 UTC 5 Oct. The other positions are the same. In addition, it is obvious that the Fitow track has great error



after adding modified CNOPs to the initial state, which indicates that the quality of the initial state is deteriorated by CNOP superimposition, and the forecasting error is increased.

Different to typhoon Fitow, all simulated tracks of typhoon Mamto are different, which are shown in Figure 9. The track data
is list in Table 8.

In the Figure 9, we can see that the typhoon Matmo tracks simulated from the WRF-ARW model are closer to the observed track by inserting the modified ADJ-CNOP to the initial state, which is only simulated by the WRF-ARW of Figure 2 (b). The reason is that there is a big difference between the WRF-ARW simulated track and the observed track. That is to say, the quality of the initial state is not very well, which results in a forecast error. The action of adding errors improves the initial
state instead, which makes the forecasting track error smaller. This result also provides a new idea for the application of CNOP method in the typhoon track simulation research.

Except the position at 1800 and 0000 UTC 21 Jul, the track data is all different, especially at 1800 UTC 22 Jul. Analysis combining with Figure 9, when all CNOP is retained, the position at 1800 UTC 22 Jul is the furthest from the observed position, and as the decreased CNOP, it moves to the observed position. The results also prove the sensibility of CNOP.
Next, the influence of the ACPW-CNOP on the simulated typhoon track is analysed. The tracks are drawn in Figures 10 and 11, and position data also is list in Tables 6 and 7. For typhoon Fitow, four sub-figures of the Figure 10 are the same, because the different ACPW-CNOPs have the same influence on the simulated typhoon tracks.

For typhoon Matmo, there are two same sub-figures, i.e. Figure 11 (b) and (c), and two different sub-figures, i.e. Figure 11 (a) and (d). And the different position is the initial position at 1800 UTC 21 Jul and the end position at 1800 UTC 22 Jul. Moreover,
the difference of data is little.

Comparing the effects of ADJ-CNOP and ACPW-CNOP on typhoon track simulation, it is found that for Fitow case, the two CNOPs have the similar effect on typhoon track, while for Matmo case, the influence of ACPW-CNOP is less than that of ADJ-CNOP. The typhoon track generated by adding ACPW-CNOP is very close to that simulated track of the WRF-ARW model (Figure 2 (b)), but much closer to the observed typhoon track.

**4.5.2 Simulated track analysis of adding CNOP modified in the sensitive regions of TTOs**

After reducing CNOPs in the sensitive regions of TTOs identified by ADJ-method and ACPW, the typhoon tracks are shown in Figures 12 and 13, which are simulated by the WRF-ARW model with adding modified CNOPs to the initial state. The strategy of reducing CNOP is change the values to 0.5 time only in the sensitive regions. The tracks data can be seen in Tables 7 and 8.
The analysis results of Figure 10 and Figure 11 are consistent with those above, that is, in Fitow case, the sensitive regions identified by ACPW-CNOP and ADJ-CNOP have the similar influence on typhoon track simulation; in Matmo case, the sensitive regions identified by ACPW-CNOP have less influence than ADJ-CNOP, but when adding the observations in the



sensitive regions, i.e. reducing the CNOP values in the sensitive regions, the simulated tracks are better than those simulated by the WRF-ARW model (Figure 2 (b)).

In conclusion, the sensitive regions identified by the ACPW-CNOP has the similar influence with the ADJ-CNOP on the simulation of typhoon tracks, sometimes the ACPW-CNOP has more positive impact on the simulation of typhoon tracks.

## 5 5 Summaries and Conclusions

In this paper, we rewrite the ACPW and applied it to solve CNOP in the WRF-ARW for identifying sensitive areas of TTOs, which is proposed by us in the study of Zhang et al. (2018), to investigate its feasibility and effectiveness in the WRF-ARW model. We take two typhoons as study cases, Fitow (2013) and Matmo (2014), and simulate them with the 60 km horizontal resolution. Similar to our previous study (Zhang et al., 2018), the total dry energy is adopted as the objective function. The

10 CNOP is also calculated by the ADJ-method as the benchmark. To evaluate the ACPW-CNOP, five aspects are analysed, such as the pattern, energy, similarity, benefits from the CNOPs reduced in the whole domain and the sensitive regions identified, and the simulated typhoon tracks.

Based on the experimental results, the following conclusions can be drawn:

(1) The temperature and wind patterns of ACPW-CNOP is similar to those of the ADJ-CNOP in all typhoons. The distribution

of temperature zones and the direction of wind vector is almost the same. Except for the CNOP patterns of ACPW are more dispersed and fragmented than those of ADJ-method.

(2) The similarity values of ADJ-CNOP and ACPW-CNOP of two typhoon cases are more than 0.5. The Fitow is 0.61, and the Matmo is 0.53.

(3) When reducing CNOPs in the whole domain, the forecasting benefits of ACPW-CNOP is greater than that of ADJ-CNOP

in typhoon Fitow, while the result is opposite in typhoon Matmo. When reducing CNOPs in the sensitive regions, the forecast income of ACPW-CNOP is greater than that of ADJ-CNOP in all typhoons.

(4) When ACPW uses the parameters in line 1 of Table 6 and the ADJ-method uses one first guess field, the speedup of the ACPW is 2.59. If the ADJ-method uses four first guess fields, the speedup of the ACPW is 10.34. In the experiments of this paper, the ACPW is more efficient than the ADJ-method.

(5) The sensitive regions identified by the ACPW-CNOP has the similar influence with the ADJ-CNOP on the simulation of typhoon tracks, sometimes the ACPW-CNOP has more positive impact on the simulation of typhoon tracks.

Overall, the feasibility and effectiveness of ACPW is proved in the WRF-ARW model.

To compared with the ADJ-method, it is limited when we construct the physical parameterization schemes of WRF-ARW. Because the corresponding adjoint model only provides one physical parameterization scheme. And that may be the reason of

30 bad simulated Fitow typhoon track. Since the ACPW method is free of the adjoint model, we will try more complicated physical parameterization schemes and improve the horizontal resolution to do such research. Moreover, ACPW can be used



to solve CNOP in the numerical models no having adjoint model, such as GFDL (Geophysical Fluid Dynamics Laboratory) and CESM (Community Earth System Model).

**Acknowledgments:** In this paper, the research was sponsored by the Fundamental Research Funds for the Central Universities of China in 2017 and the National Key Research and Development Program of China (Grant 2018YFC1506402).

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





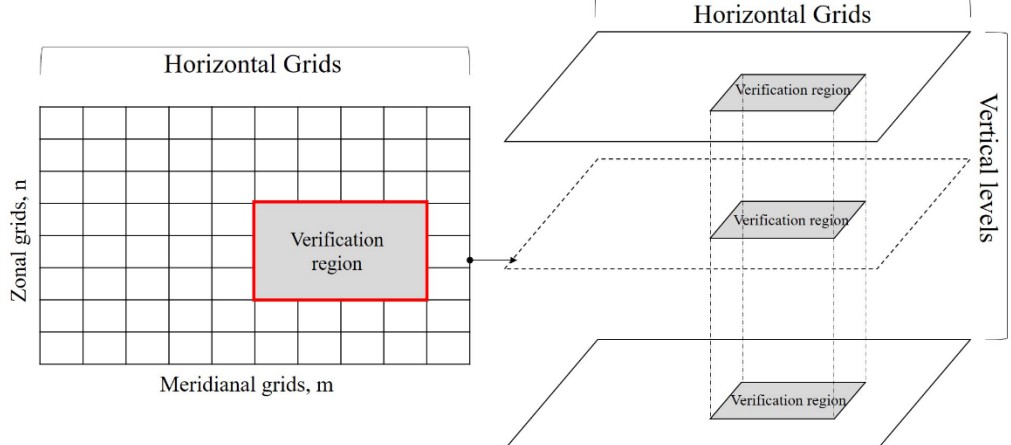

**Figure 1: The schematic diagram of verification region. The red square represents the verification region, m is the number of meridional grids and n is the number of zonal grids in horizontal grids.**

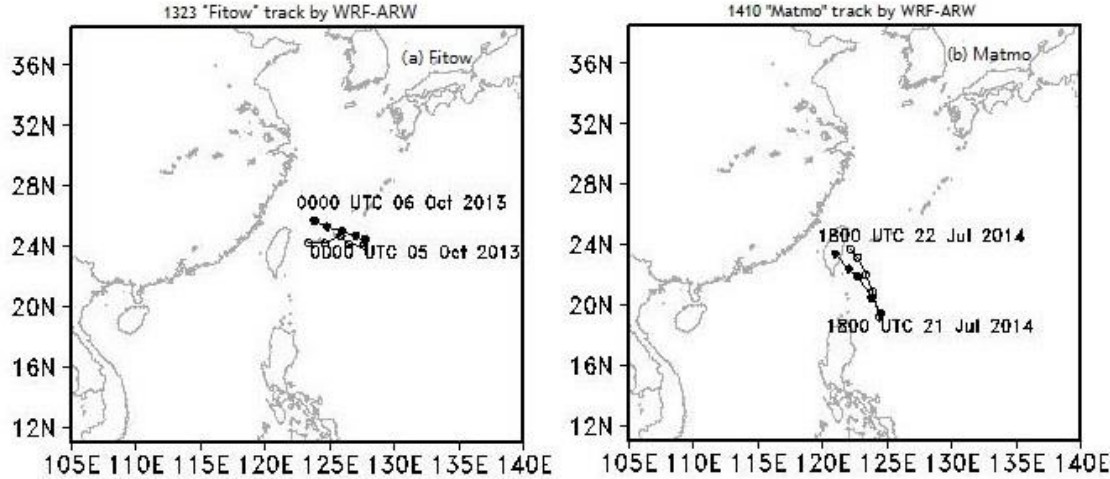

**Figure 2: Simulated tracks of WRF-ARW model and typhoon observed tracks of CMA. Solid circles represent typhoon observed tracks of CMA, hollow circles show simulated tracks. (a) Fitow from 0000 UTC 5 Oct to 0000 UTC 6 Oct 2013. (b) Matmo from 1800 UTC 21 Jul to 1800 UTC 22 Jul 214.**



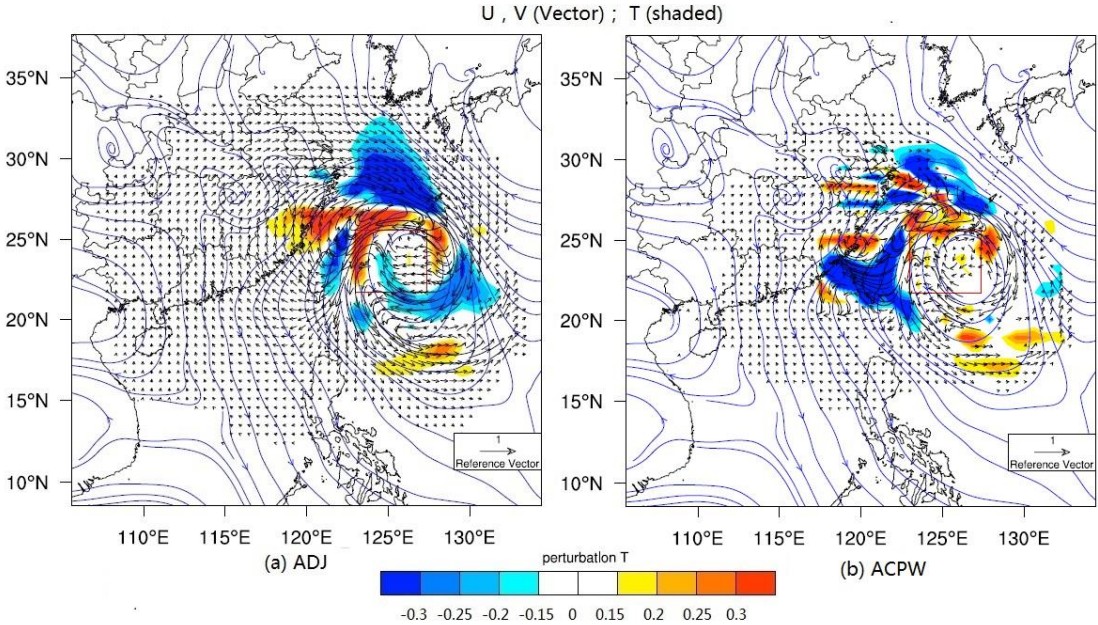

**Figure 3: The CNOP patterns of Fitow at $\eta = 0.7$. The shaded parts represent the temperature (units: K) and the vectors describe the wind (units: ms-1). The squares draw the verification areas. (a) denotes the CNOP pattern of the ADJ-method and (b) presents the ACPW.**

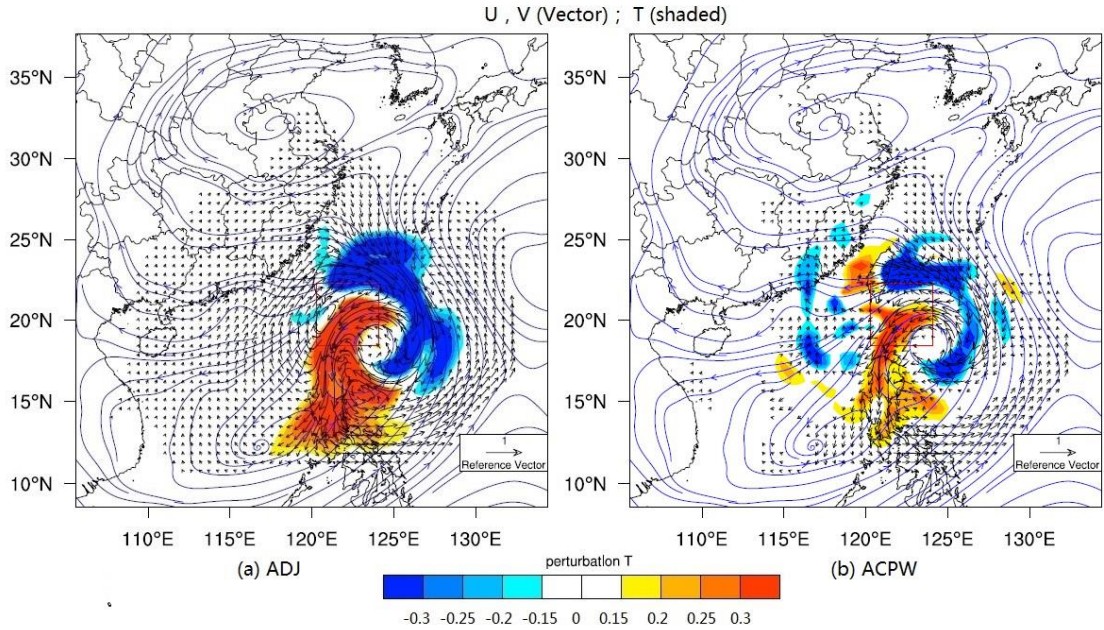

**Figure 4: The CNOP patterns of Matmo at $\eta = 0.7$. The shaded parts represent the temperature (units: K) and the vectors describe the wind (units: ms-1). The squares draw the verification areas. (a) denotes the CNOP pattern of the ADJ-method and (b) presents the ACPW.**



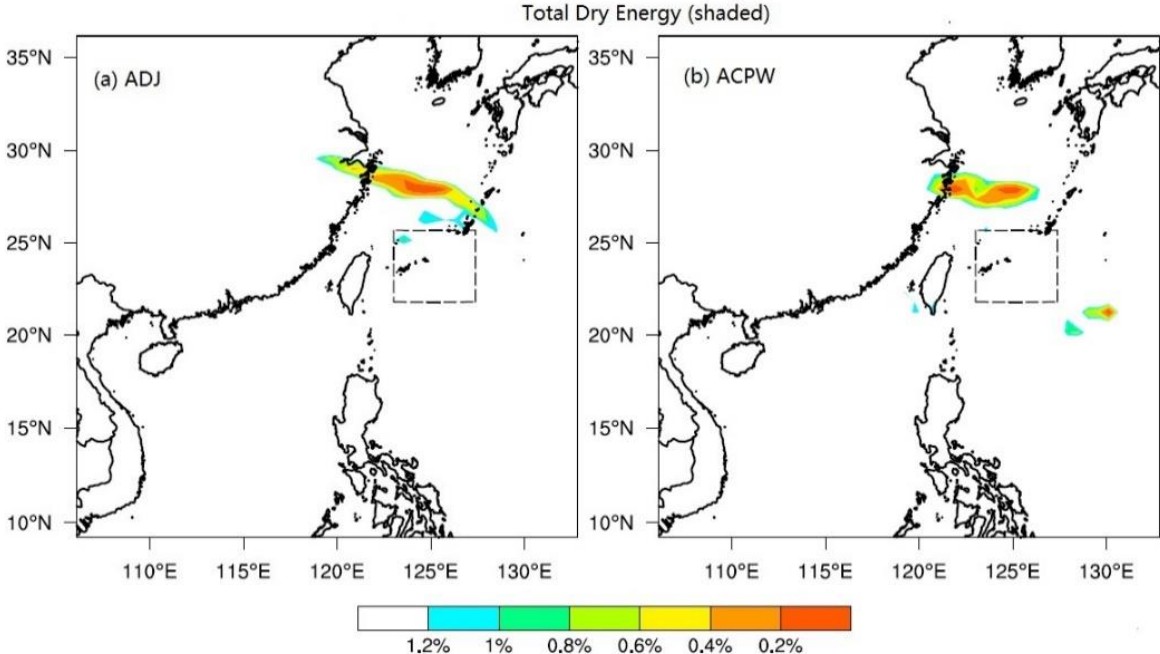

**Figure 5:** Same as Figure 3, but the shaded parts represent the vertically integrated energies for Fitow (units: J kg-1), and the black virtual box represents the validation area.

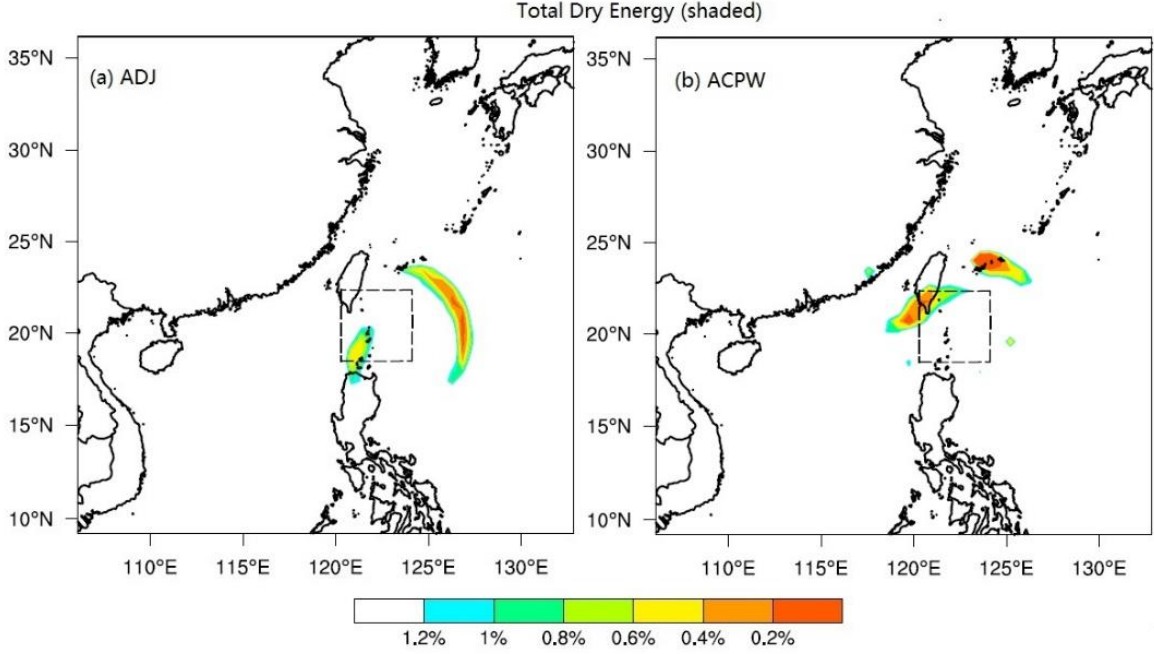

5       **Figure 6:** Same as Figure 4, but the shaded parts represent the vertically integrated energies for Fitow (units: J kg-1), and the black virtual box represents the validation area.

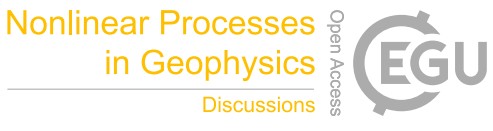

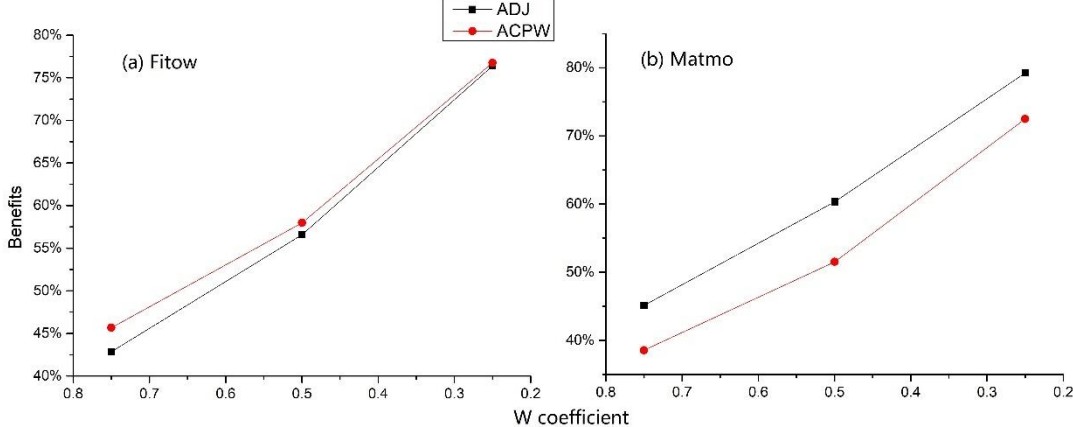

**Figure 7: Benefits (percent, %) achieved by reducing CNOPs to W×CNOPs of ADJ and ACPW methods in the whole domain for typhoon Fitow (2013). The x-coordinate is the W coefficient values. And the y-coordinate denotes the benefits (percent, %) derived from the two methods. ADJ-method is described as black line with squares and ACPW is red line with circles.**

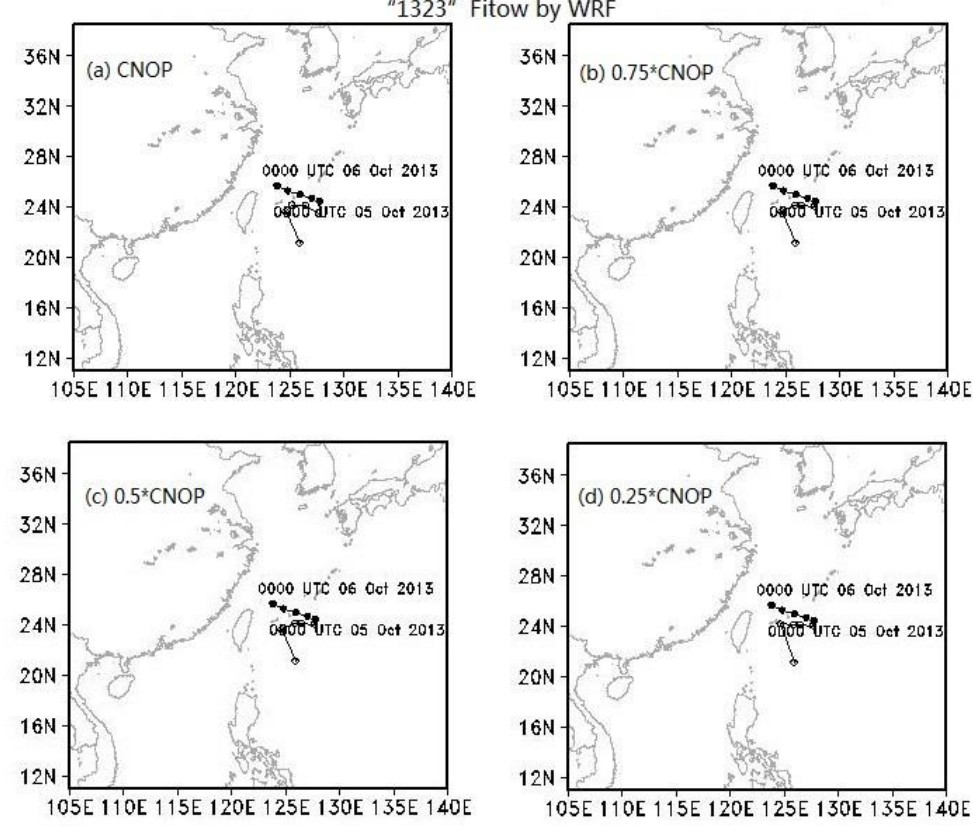

**Figure 8: Simulated typhoon tracks of WRF-ARW with adding ADJ-CNOP or W×ADJ-CNOP into the initial state in the whole domain for Fitow. Solid circles represent observed tracks of CMA, and hollow circles show the simulated tracks of the WRF-ARW model. (a), (b), (c) and (d) denote CNOP, 0.75×CNOP, 0.5×CNOP and 0.25×CNOP, respectively.**

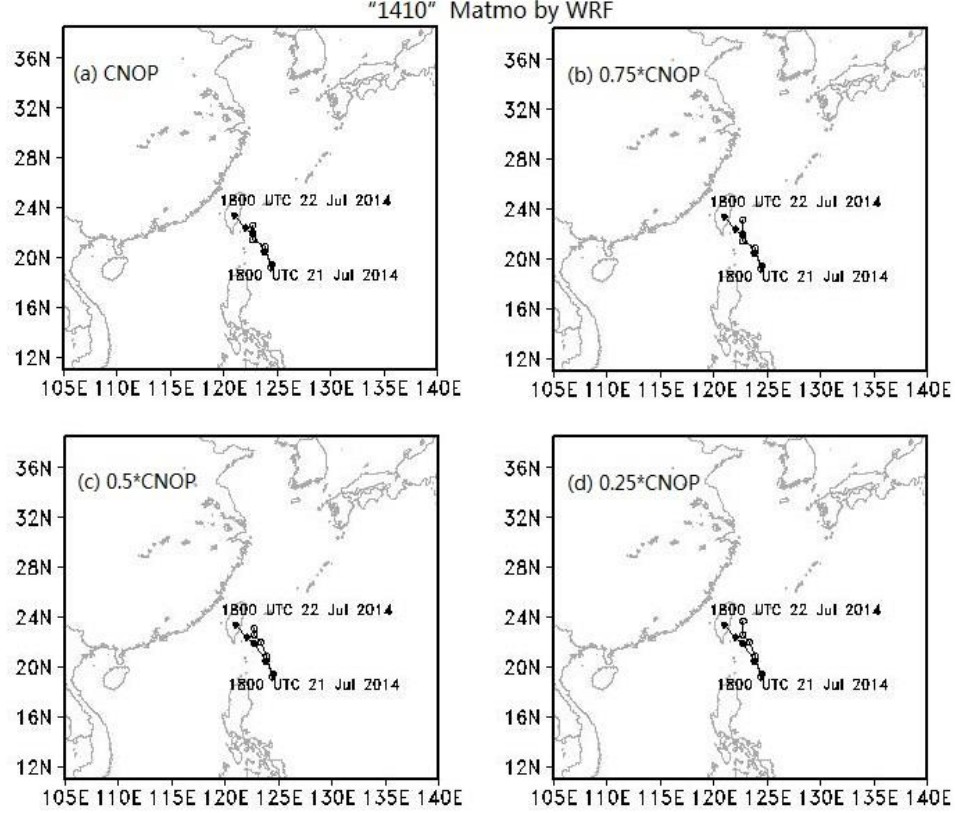

**Figure 9: Simulated typhoon tracks of WRF-ARW with adding ADJ-CNOP or W×ADJ-CNOP into the initial state in the whole domain for Matmo. Solid circles represent observed tracks of CMA, and hollow circles show the simulated tracks of the WRF-ARW model. (a), (b), (c) and (d) denote CNOP, 0.75×CNOP, 0.5×CNOP and 0.25×CNOP, respectively.**



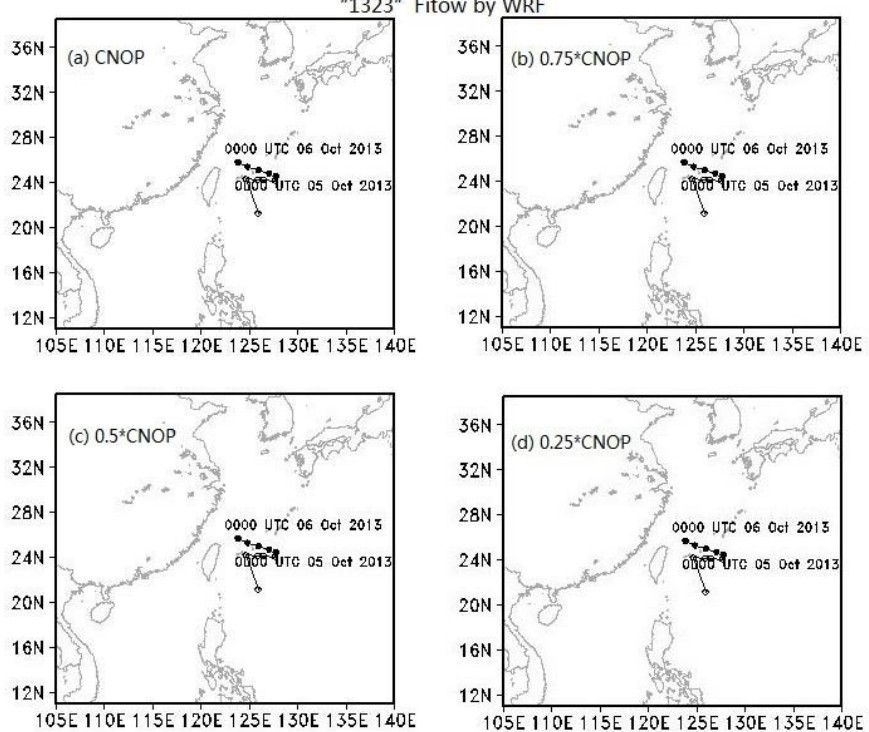

**Figure 10: Simulated typhoon tracks of WRF-ARW with adding ACPW-CNOP or W×ACPW-CNOP into the initial state in the whole domain for Fitow. Solid circles represent observed tracks of CMA, and hollow circles show the simulated tracks of the WRF-ARW model. (a), (b), (c) and (d) denote CNOP, 0.75×CNOP, 0.5×CNOP and 0.25×CNOP, respectively.**

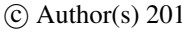


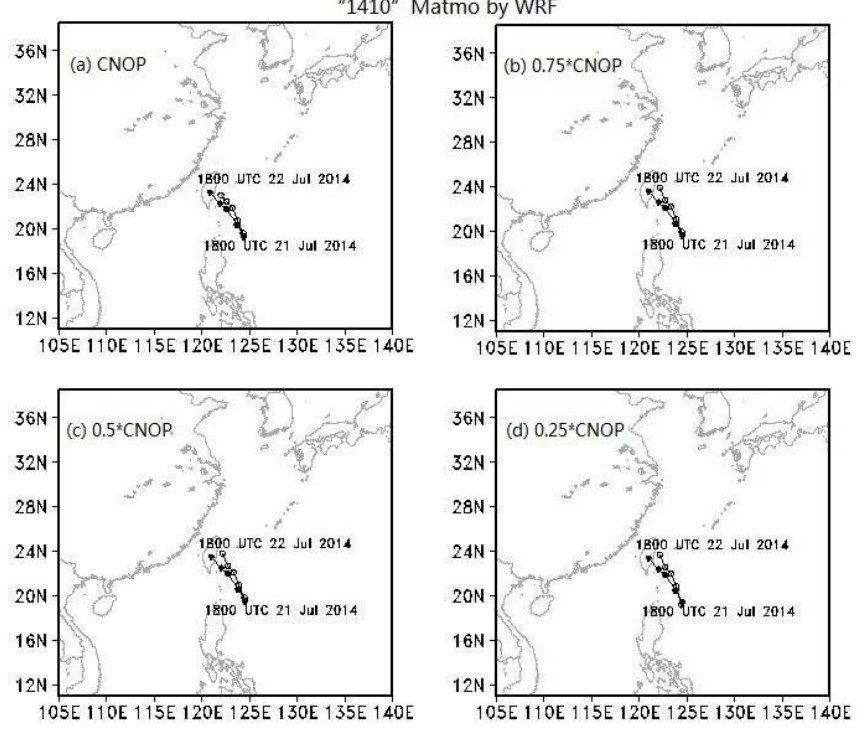

**Figure 11: Simulated typhoon tracks of WRF-ARW with adding ACPW-CNOP or W×ACPW-CNOP into the initial state in the whole domain for Matmo. Solid circles represent observed tracks of CMA, and hollow circles show the simulated tracks of the WRF-ARW model. (a), (b), (c) and (d) denote CNOP, 0.75×CNOP, 0.5×CNOP and 0.25×CNOP, respectively.**

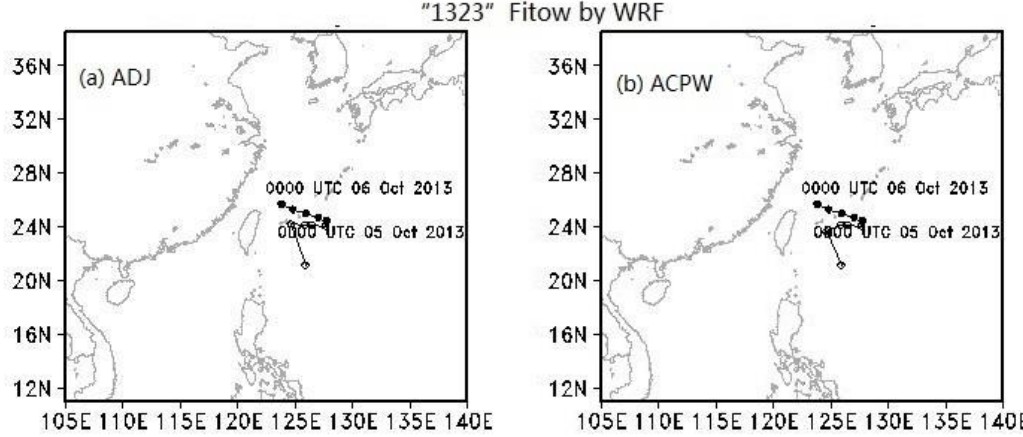

**Figure 12: Simulated Fitow tracks of the WRF-ARW with adding modified CNOPs into the initial state. Solid circles represent observed typhoon tracks of CMA, and hollow circles show the simulated typhoon tracks of the WRF-ARW model. (a) is from ADJ-method and (b) is ACPW method.**



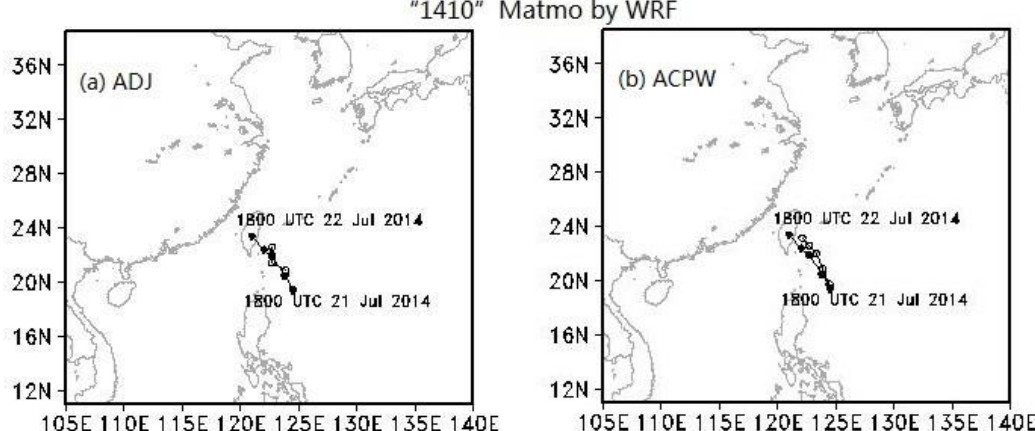

**Figure 13: Simulated Matmo tracks of the WRF-ARW with adding modified CNOPs into the initial state. Solid circles represent observed typhoon tracks of CMA, and hollow circles show the simulated typhoon tracks of the WRF-ARW model. (a) is from ADJ-method and (b) is ACPW method.**



**Table 1: The pseudocode of ACPW.**

| Algorithm. ACPW |
| --- |

Initialization:

1: Set the parameters of ACPW in Table 2.

ACPW:

2: Randomly generate an initial perturbation swarm $P_0 = \{X_1^{n'}, \ X_2^{n'}, \ \cdots, X_N^{n'}\}$, where $X^{n'} = \{x_1, x_2, \cdots, x_{n'}\}, x_i \in [-1,1], i = [1, n']$, and $n'$ is the number of principal components selected;

3: According to the adaptive coefficient α, divide the entire initial swarm $P_0$ into two subswarms $P_0^1 = \{X_1^{n'}, \ X_2^{n'}, \ \cdots, X_{N\cdot\alpha}^{n'}\}$ and $P_0^2 = \{X_{N-N\cdot\alpha}^{n'}, \ X_{N-N\cdot\alpha+1}^{n'}, \ \cdots, X_N^{n'}\}$;

4: WHILE $t < Total\_Step$ DO;

5:      $Project(X_t^{n'}, \zeta)$; Pull back the individual when it goes beyond the boundary, i.e., when $\|x_i\| > \zeta$, then $x_i = \frac{\zeta}{\|x_i\|} \times x_i$, $\zeta$ is the constrain condition in the formula (1);

6:      $AdaptFun(P_i)$; Calculate the adaptive value of the objective function parallelly, i.e., $J(x_i)$ in Eq. (1).

7:      Compare the values of the objective function of all individuals and save the best one;

8:      Calculate the difference of the best objective function values of generations $P_t$ and $P_{t-1}$,

       If the difference is smaller than the threshold value $\varepsilon$, then

         change the adaptive coefficient α to α + 0.05,

     Else

         change the adaptive coefficient α to α − 0.05,

     End if

9:      Calculate the number of two subswarms by the new adaptive coefficient α;

10:      Update the individuals of $P_t^1$ as the PSO rules;

11:      Update the individuals of $P_t^2$ as the WSA rules;

12: END WHILE

Output: CNOP

**Table 2: The control parameters of ACPW.**

| Name | Meaning | Value |
| --- | --- | --- |
| $n'$ | Number of principle components | in Table 6 |
| $N$ | Number of individuals | in Table 6 |
| $a$ | Adaptive coefficient | Initial: 0.5 |
| $\omega$ | Inertia coefficient | 0.8 |

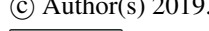
| | | |
|---|---|---|
| $c1$ | Self-awareness to track the historically optimal position | 2.05 |
| $c2$ | Social-awareness of the particle swarm to track the globally optimal position | 2.05 |
| $\Upsilon$ | Restraint factor to control the speed | 0.729 |
| $\theta$ | Velocity of individual moving | 0.5 |
| $r$ | Local optimizing radius | 8×δ/original dimensions |
| $s$ | Step size of updating individual | 0.6 |
| $p_a$ | Probability of individual escaping from current position | 0.3 |
| Total_Step | The number of iterations | in Table 6 |

Table 3: The meanings of all symbols

| Symbols | Values/ components | Meanings |
|---|---|---|
| $\zeta\varphi_0$ | $u_0', v_0', T_0', p_{s0}',$ | Initial perturbation |
| $\varphi_{NT}$ | $u_t', v_t', T_t', p_{st}'$ | Nonlinear evolution of perturbed $\Phi_0$ at time t |
| $D$ | Values rely on cases | Verification area |
| $\eta$ | (0, 1] | Vertical coordinate |
| $c_p$ | 1005.7 J kg−1 K−1 | Specific heat at constant pressure |
| $R_a$ | 287.04 J kg−1K−1 | Gas constant of dry air |
| $T_r$ | 270K | Constant parameter |
| $p_r$ | 1000hPa | Constant parameter |

Table: 4 The numerical similarity between ACPW-CNOP and ADJ-CNOP

| ACPW&ADJ | similarity |
|---|---|
| Fitow | 0.61 |
| Matmo | 0.53 |

Table 5: Benefits (percent, %) achieved by reducing CNOPs in the sensitive regions.

| Case | Method | 0.75 | 0.5 | 0.25 |
|---|---|---|---|---|
| Fitow | ADJ | 6.9169% | 6.9699% | 6.669% |
| | ACPW | **10.2871%** | **9.6823%** | **8.812%** |
| Matmo | ADJ | 0.0809% | 0.083% | 0.0779% |
| | ACPW | **2.3484%** | **3.5716%** | **5.2212%** |

Table 6: the running time of ACPW for solving CNOP in the WRF-ARW model.





| $n'$ | $N$ | *Total_Step* | Time (units: minutes) | The adaptive value of objective function |
|---|---|---|---|---|
| 30 | 20 | 10 | 89.83 | 28126.185933 |
| 30 | 40 | 10 | 179.55 | 27125.406996 |
| 30 | 60 | 10 | 269.33 | 26708.582565 |
| 30 | 40 | 20 | 359.79 | 30421.490441 |
| 40 | 30 | 10 | 136.77 | 24525.656206 |
| 40 | 40 | 20 | 360.43 | 31055.582842 |
| 50 | 30 | 10 | 136.89 | 17137.857070 |
| 60 | 30 | 10 | 137.23 | 14285.603508 |

**Table 7: The Fitow tracks of adding different CNOPs with the WRF-ARW model. The red number represents the different position at the same time point.**

| Method/Fitow (2013) | Time | CNOP | 0.75CNOP | 0.5CNOP | 0.25CNOP | 0.5Sens |
|---|---|---|---|---|---|---|
| ADJ | 0000 UTC 5 Oct | 127.618 23.5222 | 127.652 24.0799 | 127.652 24.0799 | 127.652 24.0799 | 127.652 24.0799 |
| | 0600 UTC 5 Oct | 126.43 24.1385 | 126.43 24.1385 | 126.43 24.1385 | 126.43 24.1385 | 126.43 24.1385 |
| | 1200 UTC 5 Oct | 125.207 24.187 | 125.819 24.1641 | 125.819 24.1641 | 125.819 24.1641 | 125.819 24.1641 |
| | 1800 UTC 5 Oct | 124.574 23.6492 | 124.574 23.6492 | 124.574 23.6492 | 124.595 24.2075 | 124.595 24.2075 |
| | 0000 UTC 6 Oct | 125.874 21.1697 | 125.874 21.1697 | 125.874 21.1697 | 125.874 21.1697 | 125.874 21.1697 |
| ACPW | 0000 UTC 5 Oct | 127.652 24.0799 | 127.652 24.0799 | 127.652 24.0799 | 127.652 24.0799 | 127.652 24.0799 |
| | 0600 UTC 5 Oct | 126.43 24.1385 | 126.43 24.1385 | 126.43 24.1385 | 126.43 24.1385 | 126.43 24.1385 |
| | 1200 UTC 5 Oct | 125.819 24.1641 | 125.819 24.1641 | 125.819 24.1641 | 125.819 24.1641 | 125.819 24.1641 |
| | 1800 UTC 5 Oct | 124.595 24.2075 | 124.595 24.2075 | 124.595 24.2075 | 124.595 24.2075 | 124.574 23.6492 |



| | 0000 UTC 6 | 125.874 | 125.874 | 125.874 | 125.874 | 125.874 |
| | Oct | 21.1697 | 21.1697 | 21.1697 | 21.1697 | 21.1697 |

**Table 8: The Matmo tracks of adding different ADJ-CNOPs with the WRF-ARW model. The red number represents the different position at the same time point.**

| Method/Matmo (2014) | Time | CNOP | 0.75CNOP | 0.5CNOP | 0.25CNOP | 0.5Sens |
|---|---|---|---|---|---|---|
| ADJ | 1800 UTC | 124.413 | 124.413 | 124.413 | 124.413 | 124.413 |
| | 21 Jul | 19.193 | 19.193 | 19.193 | 19.193 | 19.193 |
| | 0000 UTC | 123.876 | 123.876 | 123.876 | 123.876 | 123.876 |
| | 22 Jul | 20.8786 | 20.8786 | 20.8786 | 20.8786 | 20.8786 |
| | 0600 UTC | 122.696 | 122.696 | 123.31 | 123.31 | 122.696 |
| | 22 Jul | 21.4631 | 21.4631 | 22.0081 | 22.0081 | 21.4631 |
| | 1200 UTC | 122.708 | 122.708 | 122.72 | 122.72 | 122.708 |
| | 22 Jul | 22.0207 | 22.0207 | 22.5785 | 22.5785 | 22.0207 |
| | 1800 UTC | 122.72 | 122.733 | 122.733 | 122.745 | 122.72 |
| | 22 Jul | 22.5785 | 23.1367 | 23.13 | 23.695 | 22.5785 |
| ACPW | 1800 UTC | 124.433 | 124.433 | 124.433 | 124.413 | 124.433 |
| | 21 Jul | 19.7486 | 19.7486 | 19.7486 | 19.193 | 19.7486 |
| | 0000 UTC | 123.876 | 123.876 | 123.876 | 123.876 | 123.876 |
| | 22 Jul | 20.8786 | 20.8786 | 20.8786 | 20.8786 | 20.8786 |
| | 0600 UTC | 123.31 | 123.31 | 123.31 | 123.31 | 123.31 |
| | 22 Jul | 22.0081 | 22.0081 | 22.0081 | 22.0081 | 22.0081 |
| | 1200 UTC | 122.72 | 122.72 | 122.72 | 122.72 | 122.72 |
| | 22 Jul | 22.5785 | 22.5785 | 22.5785 | 22.5785 | 22.5785 |
| | 1800 UTC | 122.125 | 122.135 | 122.135 | 122.135 | 122.125 |
| | 22 Jul | 23.1468 | 23.7051 | 23.7051 | 23.7051 | 23.1468 |