# Peer review of "CNOP based on ACPW for Identifying Sensitive Regions of Typhoon Target Observations with WRF Model"

_Nonlinear Processes in Geophysics, 2019_

## Referee Comment (RC1) · Anonymous Referee #1 · 15 Jun 2019

this paper propsed something interesting, however, the paper is not clear in current version, and its language also requires improvement. In summary, there are several comments that may help to improve the quality of this manuscript.

1) The english requires improvement, some sentences look like just the direct translation of Chinese, it is prefered to let a native speaker check the whole language style of the manuscript.

2) In the proposed work, the authors only use CMA trace data for verification, which is considered not solid and strong. I suggest the authors to use more trace data for analysis and comparison, such as trace data provided by JMA and USA.

[Figure]

3) The authors should explain the reason why these two typhoons are condiered for analysis in this work, why they are representitive? In fact, for further verfiication, more typhoon cases should be considered.

4) In the conclusion, the similarity of ADJ-CNOP and ACPW-CNOP of two typhoons are not very large.Or in other words, the similarity calculation is just cross-correlation.

5) Some terms are not clear at all, for general readers, such as the first guess field, the speed up and some others, the author should explain those terms for better understanding.

6) The author should explain the importance of adoint model and explain whether it is true necessary during the algorithm realization.

7) Why all the dry energy are adopted as objective function?

---

## Referee Comment (RC2) · Anonymous Referee #2 · 26 Jun 2019

In the present paper the authors apply the ACPW (see paper for the meaning of this acronym) algorithm to the WRF-ARW model to investigate its feasibility and effectiveness. ACPW has been been proposed by (almost) the same authors (Zhang et al. 2018). Here, the authors (basically) repeat the simulations and analysis of Zhang et al. using WRF-ARW instead of MM5. As in Zhang et al. two typhoons (Fitow and Matmo) serve as a testbed. The results are very similar to that in Zhang et al. (2018) indicating that ACPW can indeed been applied to WRF-ARW too.

General Introducing and evaluating new methods to improve the prediction (or our understanding) of tropical storms are valuable contributions. In my view, however, the

present study does not add (much) to prediction nor understanding beyond Zhang et al. The only new aspect appears that ACPW algorithm may also be applied to other models than MM5, which could be somehow expected. Therefore, unfortunately, I cannot recommend publication. Please note that this does not mean that I disregard the technical efforts to adapt the algorithm to a new model.

Some additional major comments:

- The English needs substantial improvements.

- At the moment the paper reads like an adapted/modified and shortened version of Zhang et al., without trying to get some 'added value'. Furthermore, at some places the meaning is not clear without the Zhang et al. paper (e.g. the definition of 'forecasting benefits' (Chapter 4.3.1), the pseudocode (Table 1 in particular 8,10,11)).

- As the authors note (P2 L10ff, P10L28), the adjoint version of WRF-ARW used for this study appears not very well suited for the present purpose (typhoon prediction). It is not clear how important this issue is for the conclusions drawn by the authors.

---

## Referee Comment (RC3) · Anonymous Referee #3 · 1 Jul 2019

[12pt]article  epsfig, amsmath

**Review of "CNOP based on ACPW for identifying... WRF model**
**by Mu et al."**

The paper describes an algorithm –ACPW– to compute conditional nonlinear optimal perturbation –CNOP– using the WRF–ARW model to identify sensitive areas of typhoon-target observations. The authors apply it to two cases –Filow and Matmo. Results are based on maximizing the total dry energy. They then compare their results with those obtained using the adjoint model algorithm.

The authors conclude that the ACPW provides over all better results than the adjoint algorithm, particularly in the sensitive regions, and is more efficient.

Recommendation

Reject and resubmit.

Although the idea put forward in the paper is good the writing really needs attention. Besides, I find that the notation related to the equations is not proper.
I was caught between major revision and reject/resubmit. But it seems that the paper needs major rewriting and also need to be checked by a native speaker.

Major concern

*Equations and notation*
Starting with the line 10, pg3, – a perturbation of a quantity $\varphi$ is conventionally noted $\delta\varphi$ (like $\varphi$'), where $\delta$ is understood to be an operator. The notation $\zeta\varphi_0$ is misleading. In addition, $\delta\varphi_0$ of $\varphi_0$ not $\Phi_0$.
Also requring $\|\varphi_0\|^2 \leq \zeta$ ? $\zeta$ is an operator in the text and now it is like a number?

The costfunction $J$ is introduced in top of pg 3, but only explained and detailed 2 pages later?

P: projection operator – what kind of projection, and on which space?

$\Phi_t$ (should be $\varphi_t$ for consistency) is not an operator – it is the state of the system at

l18: CNOP is an optimization algorithm and not a cost-function
l23: environment idealized ??? Forecast income ???
Time consumption: CPU time.

*Content*

1. Above all, it is not clear what is the main difference with Zhang et al. (2108), and what is the advantage of the new algorithm. Any concrete results ?

2. The authors use PCs to reduce the problem dimension. It is not clear how the PCs are obtained: PCs of what, and what is the sample size used to get these PCs? Are the authors using the 24-hr data with 6-hr sampling?

3. Not clear how is the sensitive region determined as CNOP only identifies initial perturbations. Are the authors computing the costfunction for different regions then compare them?

**Supplement:**

**Review of "CNOP based on ACPW for identifying... WRF model by Mu et al."**

The paper describes an algorithm –ACPW– to compute conditional nonlinear optimal perturbation –CNOP– using the WRF–ARW model to identify sensitive areas of typhoon-target observations. The authors apply it to two cases –Filow and Matmo. Results are based on maximizing the total dry energy. They then compare their results with those obtained using the adjoint model algorithm.

The authors conclude that the ACPW provides over all better results than the adjoint algorithm, particularly in the sensitive regions, and is more efficient.

**Recommendation**

Reject and resubmit.

Although the idea put forward in the paper is good the writing really needs attention. Besides, I find that the notation related to the equations is not proper.

I was caught between major revision and reject/resubmit. But it seems that the paper needs major rewriting and also need to be checked by a native speaker.

**Major concern**

*Equations and notation*

Starting with the line 10, pg3, – a perturbation of a quantity $\varphi$ is conventionally noted $\delta\varphi$ (like $\varphi$'), where $\delta$ is understood to be an operator. The notation $\zeta\varphi_0$ is misleading. In addition, $\delta\varphi_0$ of $\varphi_0$ not $\Phi_0$.

Also requring $\|\varphi_0\|^2 \leq \zeta$ ? $\zeta$ is an operator in the text and now it is like a number?

The costfunction $J$ is introduced in top of pg 3, but only explained and detailed 2 pages later?

P: projection operator – what kind of projection, and on which space?

$\Phi_t$ (should be $\varphi_t$ for consistency) is not an operator – it is the state of the system at time t.

Eq (4): Please describes all the parameters

Eq (5): the third '+' should be '-'

Pg 6, l18 - W is not introduced before.

*Writing/English*

The writing really needs care all across the entire manuscript. I only give some examples

below.

Pg 3, l25: is –> are

l26: list –> listed

l26: Delete the secon Zhange et al.

Pg 4, l23: "are checked built on" ??

l29: meaning not clear

Pg 5, l20: is –> are

l21: last sentence not clear

l23: total vertical –> vertically integrated

Pg 6, l1: last sentence: rewrite.

l3: "distribution of ... then" Rewrite, and distribution is not the right word.

Section 4.2 "numerical similarity" –> spatial correlation

Section 4.3 'Benefits" ???

l18: CNOP is an optimization algorithm and not a cost-function

l23: environment idealized ??? Forecast income ???

Time consumption: CPU time.

*Content*

1. Above all, it is not clear what is the main difference with Zhang et al. (2108), and what is the advantage of the new algorithm. Any concrete results ?

    2. The authors use PCs to reduce the problem dimension. It is not clear how the PCs are obtained: PCs of what, and what is the sample size used to get these PCs? Are the authors using the 24-hr data with 6-hr sampling?

    3. Not clear how is the sensitive region determined as CNOP only identifies initial per-turbations. Are the authors computing the costfunction for different regions then compare them?

---

## Author Comment (AC1) · 15 Aug 2019

List of Responses Responds to the Anonymous Referee #1's comments: Special thanks for your good comments which are very useful for us to improve the paper. 1. Response to comment: this paper proposed something interesting, however, the paper is not clear in current version, and its language also requires improvement. In summary, there are several comments that may help to improve the quality of this manuscript. 1) The english requires improvement, some sentences look like just the direct translation of Chinese, it is preferred to let a native speaker check the whole language style of the manuscript. Response: As Reviewer1 suggested that we have tried

our best to improve the presentation of this paper, and correct the syntax and spelling errors.

2. Response to comment: In the proposed work, the authors only use CMA trace data for verification, which is considered not solid and strong. I suggest the authors to use more trace data for analysis and comparison, such as trace data provided by JMA and USA. Response: Thanks the advice of Reviewer1, but the CMA trace data is enough for the experimental design of this paper. Because the current experiments are under the ideal conditions. As your suggestion, we will use trace data provided by JMA and USA to do the further research about real environment. 3. Response to comment: The authors should explain the reason why these two typhoons are considered for analysis in this work, why they are representitive? In fact, for further verification, more typhoon cases should be considered. Response: For this paper, the experiments of two typhoons can be compared with our previous researches of MM5 model, and that can be used to prove the portability and effectiveness of ACPW method. 4. Response to comment: In the conclusion, the similarity of ADJ-CNOP and ACPW-CNOP of two typhoons are not very large. Or in other words, the similarity calculation is just cross-correlation. Response: In this paper, the purpose of solving CNOP is to identify sensitive areas of typhoon target observations. The similarity of the ADJ-CNOP and ACPW-CNOP is not the main indicator. The identified sensitive region and its influence on the forecast skills are the most important indicators. And in our paper, the influence of the different sensitive areas identified by the ADJ-CNOP and ACPW-CNOP is almost the same. 5. Response to comment: Some terms are not clear at all, for general readers, such as the first guess field, the speed up and some others, the author should explain those terms for better understanding. Response: As Reviewer1 suggested that we have explained the first guess field, the speed up and other terms in the paper for better understanding. "As the SPG2 needs to execute several times to find the best result, several different initial perturbations (also called first guess fields) are needed. In this paper, we use four first guess fields. And when we use the four first guess fields, the time consumption of ADJ method is up to 929.24

minutes." "The time consumption of ADJ method divided by the time consumption of ACPW is the speedup of the ACPW." 6. Response to comment: The author should explain the importance of adjoint model and explain whether it is true necessary during the algorithm realization. Response: As Reviewer1 suggested that we introduced that why the adjoint model is necessary in the SPG2 algorithm and why we develop the algorithm of free adjoint model. "As we all know that the SPG2 algorithm must use the adjoint model to obtain the gradient information for updating the search direction. But the adjoint model of WRF-ARW only has one gravity dragging boundary layer parameterization scheme for such study, which limits the simulation of typhoon. In addition, when the horizontal resolution is higher than 30km, the gradient information calculated by the adjoint model has errors and omissions, which results in falling into the local optimum or optimization failure." 7. Response to comment: Why all the dry energy are adopted as objective function? Response: As Reviewer1 suggested that we explained that the reason of adopting all the dry energy as objective function. "Zhou and Zhang (2014) studied three sensitive area identification schemes and recommended the vertically integrated energy scheme."

Please also note the supplement to this comment:
https://www.nonlin-processes-geophys-discuss.net/npg-2019-24/npg-2019-24-AC1-supplement.zip

---

## Author Comment (AC2) · 15 Aug 2019

List of Responses Responds to the Anonymous Referee #2's comments: Special thanks for your good comments which are very useful for us to improve the paper.

1. Response to comment: - The English needs substantial improvements. Response: As Reviewer1 suggested that we have tried our best to improve the presentation of this paper, and correct the syntax and spelling errors. 2. Response to comment: - At the moment the paper reads like an adapted/modified and shortened version of Zhang et al., without trying to get some 'added value'. Furthermore, at some places the meaning is not clear without the Zhang et al. paper (e.g.

the definition of 'forecasting benefits' (Chapter 4.3.1), the pseudocode (Table 1 in particular 8,10,11)). Response: It is really true as Rreview2 suggested that we should give more details about ACPW algorithm and other terms. "The update rules of the PSO and WSA are descripted in the following. The PSO use the classical formula (4) to update the individuals. {âŰĹ(v_iˆ(k+1)=ãĂŰ$\omega$vãĂŮ_iˆk+c_1 $\alpha$(o_iˆk-u_iˆk )+c_2 $\beta$(o_gˆk-u_iˆk)@u_iˆ(k+1)=u_iˆk+$\gamma$v_iˆ(k+1) )âŤď (4) where the superscript k is the current iteration and k+1 is the next iterative step. v_iˆ(k+1) is the updating velocity of the individual u_iˆk. $\omega$ is the inertia coefficient. c_1 is the learning factor for self-awareness to track the historically optimal position, and c_2 for social-awareness of the particle swarm to track the globally. $\alpha$ and $\beta$ are the random numbers uniformly distributing in (0, 1). o_iˆk is the local optimum and o_gˆk is the global optimum in the kˆth iteration. $\gamma$ is the restraint factor to control the speed. u_iˆ(k+1) is the updated individual. There are two ways for updating individual in WSA, prey and escape, which represent the functions of searching in a local region and escaping from a local optimum. uik+1=uik+$\theta$·r·randâňŽ distuik,uik+1<r&&.and.J(uik)<J(uik+1)uik+1=uik+$\theta$·s·escapeâňŽ p>p_a (5) where the superscript k or k+1 is also the iterative step, $\theta$ is the velocity, r is the local optimizing radius, which smaller than the global constraint radius $\delta$. rand() is the random function, whose mean value distributed in [-1,1]. escape () is the function of calculating a random position, which is larger 3 times than r. s is the step size of the updating individual. p is a random number in [0,1], p_a is the probability of individual escaping from the current position." 3. Response to comment: - As the authors note (P2 L10ff, P10L28), the adjoint version of WRF-ARW used for this study appears not very well suited for the present purpose (typhoon prediction). It is not clear how important this issue is for the conclusions drawn by the authors. Response: In this paper, the purpose of solving CNOP is to identify sensitive areas of typhoon target observations. The sensitive areas are used to improve the forecast skills. To evaluate the ACPW, we need compare it with the classical method, i.e. ADJ method. And the ADJ method must use the adjoint model. Hence, we also use the adjoint model. "Recently, there

is only one study which identify sensitive areas by using the WRF-ARW model (Yu et al., 2017). Yu et al. (2017) use the SPG2 (spectral projection gradient 2) algorithm (Ernesto et al., 2001) to solve CNOP. As we all know that the SPG2 algorithm must use the adjoint model to obtain the gradient information for updating the search direction. But the adjoint model of WRF-ARW only has one gravity dragging boundary layer parameterization scheme for such study, which limits the simulation of typhoon. In addition, when the horizontal resolution is higher than 30km, the gradient information calculated by the adjoint model has errors and omissions, which results in falling into the local optimum or optimization failure. Hence, an algorithm without using the adjoint model is needed." "To compare with the ADJ method, it is limited when we construct the physical parameterization schemes of WRF-ARW. Because the corresponding adjoint model only provides one physical parameterization scheme. And that may be the reason of bad simulated Fitow typhoon track. Since the ACPW method is free of the adjoint model, we will try more complicated physical parameterization schemes and improve the horizontal resolution to do such research. Moreover, ACPW can be used to solve CNOP in the numerical models no having adjoint model, such as GFDL (Geophysical Fluid Dynamics Laboratory) and CESM (Community Earth System Model). "

Please also note the supplement to this comment:
https://www.nonlin-processes-geophys-discuss.net/npg-2019-24/npg-2019-24-AC2-supplement.zip
* * *

---

## Author Comment (AC3) · 15 Aug 2019

List of Responses Responds to the Anonymous Referee #3's comments: Special thanks for your good comments which are very useful for us to improve the paper. 1. Response to comment: Although the idea put forward in the paper is good the writing really needs attention. Besides, I find that the notation related to the equations is not proper. I was caught between major revision and reject/resubmit. But it seems that the paper needs major rewriting and also need to be checked by a native speaker. Response: As Reviewer3 suggested that we have tried our best to improve the presentation of this paper, and correct the syntax and spelling errors. 2.

Response to comment: Starting with the line 10, pg3, – a perturbation of a quantity $\varphi$ is conventionally noted $\delta\varphi$ (like $\varphi\tilde{}$), where $\delta$ is understood to be an operator. The notation $\zeta\varphi\_0$ is misleading. In addition, $\delta\varphi\_0$ of $\varphi\_0$ not $\Phi\_0$. Also requring ‖$\varphi\_0$ ‖ˆ2≤$\zeta$ ? $\zeta$ is an operator in the text and now it is like a number? Response: As Reviewer3 mentioned, we have explained the formulas of CNOP in more detail. "where $\zeta$ is a constrained radius of an initial perturbation $\varphi\_0$ We use $\zeta\varphi\_0$ to represent the constrained initial perturbation. $\Phi\_0$ is an initial basic state and also a background field of a nonlinear numerical model. $\zeta\varphi\_0$ is a type initial perturbation, which can be insert into the initial basic state $\Phi\_0$ P is a local numerical projection operator with setting 1 inside of the verification region and 0 outside, which is an operation of matrix multiplication. And the verification region is a key area considered by researchers, which is represented in Fig. 1. M denotes a nonlinear numerical model. " 3. Response to comment: The costfunction J is introduced in top of pg 3, but only explained and detailed 2 pages later? Response: As Reviewer3 mentioned, we have explained why we introduce the detail of J in the section 3. "J denotes the objective function of solving CNOP, and the detail of its computation is described in Section 3. Because, the detail of calculating J would be different as the experimental design is different." 4. Response to comment: P: projection operator – what kind of projection, and on which space? Response: As Reviewer3 mentioned, we have explained P in more detail. "P is a local numerical projection operator with setting 1 inside of the verification region and 0 outside, which isan operation of matrix multiplication." 5. Response to comment: $\Phi\_t$ (should be $\varphi\_t$ for consistency) is not an operator – it is the state of the system at l18: CNOP is an optimization algorithm and not a cost-function Response: As Reviewer3 mentioned, we have modified the description of $\Phi\_t$." $\Phi\_t$ is the state of nonlinear evolution of M from the initial time $t\_0$ to the predicted time t." And we have explained the formula of CNOP "Combined with the formula (2), the formula (1) means that the CNOP is the initial perturbation having largest nonlinear development, i.e. J(‖$\zeta\varphi$‗\_0ˆ\* ), when it is inserted into $\Phi\_0$ and M evolves from the initial time $t\_0$ to the predicted time t with the modified initial state ($\Phi\_0+\zeta\varphi\_0$)."

In this paper, CNOP is not an optimization algorithm neither a cost-function. 6. Response to comment: l23: environment idealized ??? Forecast income ??? Time consumption: CPU time. Response: As Reviewer3 mentioned, we have modified the unsuitable words, such as income and running time. "environment idealized" is a type of assumption. "All experiments are based on two assumptions that: a. When adding target observations in the identified sensitive areas, the environment around is idealized, and the improvements of observations added are reducing original errors to 0.75, 0.5 and 0.25 times. b. CNOPs achieved by us can be seen as the optimal initial perturbations. Once we reduce them in the sensitive regions, the benefits earned will be the best. As mentioned in the subsection 4.1, the sensitive region is determined according to the first 1.2% of total vertical dry energy, as shown in Figures 5 and 6, which are the shadow zones in the figures." "Time consumption" is the time of ACPW solving CNOP. 7. Response to comment: Above all, it is not clear what is the main difference with Zhang et al. (2108), and what is the advantage of the new algorithm. Any concrete results ? Response: In this paper, we rewrite the ACPW and applied it to solve CNOP in the WRF-ARW for identifying sensitive areas of typhoon target observations. ACPW was proposed by us in 2018, the main difference between this paper and Zhang et al. (2018) is the different nonlinear model. 8. Response to comment: The authors use PCs to reduce the problem dimension. It is not clear how the PCs are obtained: PCs of what, and what is the sample size used to get these PCs? Are the authors using the 24-hr data with 6-hr sampling? Response: "Eight group of control parameters and the experimental results are list in Table 6. For the experimental analysis, the number of principal components (PCs), which are selected dimensions of the feature space from the dimension reduction of Principal Component Analysis (PCA), has little effect on the time consumption, but has great influence on the adaptive value of objective function. The samples of PCA are from the difference of the different forecast states in the forecast time. In the WRF-ARW model, we get 551 samples and reduce the dimension from 2.5*105 to 30-60 with PCA." 9. Response to comment: Not clear how is the sensitive region determined as CNOP only identifies
initial perturbations. Are the authors computing the costfunction for different regions then compare them? Response: "As we use the total vertical dry energy to identify the sensitive regions of typhoons, the distribution of the vertical dry energy is presented in Figure 5 and Figure 6. And the figures show the area with the first 1.2% energy. " "The experiments of this section include two parts: the forecasting benefits obtained by reducing CNOP to W × CNOP in the whole domain, i.e. the CNOP values of all grid points are reduced; the forecasting benefits from CNOP to W × CNOP is reduced only in the sensitive regions, i.e. the CNOP values of the sensitive grid points are reduced to 0.75 × CNOP, 0.5 × CNOP and 0.25 × CNOP. " "In order to investigate the validity of CNOP in identifying sensitive regions, we compare the 24-hour simulated typhoon track by adding CNOP or W × CNOP to the initial states. Similar to the benefits, there are two ways to modify the CNOP value: one is to reduce the CNOP value to 0.75, 0.5 and 0.25 times in the whole domain; the other is to reduce the CNOP value to 0.5 times only in the sensitive regions of TTOs. "

Please also note the supplement to this comment:
https://www.nonlin-processes-geophys-discuss.net/npg-2019-24/npg-2019-24-AC3-supplement.zip